# Chemometric Investigation and Antimicrobial Activity of *Salvia rosmarinus* Spenn Essential Oils

**DOI:** 10.3390/molecules27092914

**Published:** 2022-05-03

**Authors:** Saoussan Annemer, Abdellah Farah, Hamide Stambouli, Amine Assouguem, Mikhlid H. Almutairi, Amany A. Sayed, Ilaria Peluso, Taoufik Bouayoun, Nehal Ahmed Talaat Nouh, Abdelhakim El Ouali Lalami, Yassine Ez zoubi

**Affiliations:** 1Laboratory of Applied Organic Chemistry, Faculty of Sciences and Techniques, University Sidi Mohammed Ben Abdellah, B.P. 2202, Fes 30000, Morocco; saoussan.annemer@usmba.ac.ma (S.A.); farah.abdellah1@gmail.com (A.F.); eloualilalami@yahoo.fr (A.E.O.L.); y.ezzoubi@uae.ac.ma (Y.E.z.); 2Institute of Forensic Sciences of Gendarmerie Royal, B.P. 6597, Rabat 10000, Morocco; stambouli58@gmail.com (H.S.); bouayoun.taoufik@gmail.com (T.B.); 3Department of Zoology, College of Science, King Saud University, P.O. Box 2455, Riyadh 11451, Saudi Arabia; malmutari@ksu.edu.sa; 4Zoology Department, Faculty of Science, Cairo University, Giza 12613, Egypt; amanyasayed@sci.cu.edu.eg; 5Research Centre for Food and Nutrition, Council for Agricultural Research and Economics (CREA-AN), 00178 Rome, Italy; i.peluso@tiscali.it; 6Inpatient Pharmacy, Mansoura University Hospital, Mansoura 35516, Egypt; microbiology6.jed@bmc.edu.sa; 7Higher Institute of Nursing Professions and Health Techniques, Regional Health Directorate, EL Ghassani Hospital, Fez 30000, Morocco; 8Biotechnology, Environmental Technology and Valorization of Bio-Resources Team, Department of Biology, Faculty of Sciences and Techniques Al-Hoceima, Abdelmalek Essaadi University, Tetouan 93000, Morocco

**Keywords:** antibacterial activity, chemical composition, chemometric investigation, hierarchical cluster analysis, principal components analysis, *Salvia rosmarinus* Spenn

## Abstract

To ensure the better production and sustainable management of natural resources, a chemometric investigation was conducted to examine the effect of cooperative and harvesting periods on the crop yields and chemical compositions of *Salvia rosmarinus* Spenn essential oils in the Oriental region of Morocco. The samples were collected from three cooperatives over nine time periods from January 2018 to April 2019. The chemical composition of *Salvia rosmarinus* Spenn essential oils was analyzed by gas chromatography coupled with mass spectrometry. The data from this study were processed by multivariate analyses, including principal component analysis (PCA) and hierarchical cluster analysis (HCA). The disc diffusion technique and a determination of the minimal inhibitory concentration were performed to study the antibacterial properties of the oils. Statistical analysis showed that the cooperative and harvest period have a significant effect on yields. The highest yield of essential oil was recorded in April 2019 at cooperative C1. The PCA and the HCA results were divided into two groups: Group A for the summer season and group B for the winter season. The samples collected during summer were characterized by a high amount of 1,8-cineole component and a high yield of essential oil, whereas the samples collected during winter were qualified by a high amount of α-pinene component and a low yield of essential oil. The antibacterial activity of *Salvia rosmarinus* Spenn essential oils showed that *Mycobacterium smegmatis* ATCC23857 and *Bacillus subtilis* ATCC 23857 are the most susceptible strains, stopping growth at 1/500 (*v*/*v*). The least susceptible strain is Escherichia coli ATCC25922, with an MIC value corresponding to 1/250 (*v*/*v*). The findings of this study could have a positive economic impact on the exploitation of rosemary in the Oriental region, especially during the best harvest periods, as they indicate how to obtain the best yields of oils richest in 1,8-cineole and α-pinene chemotypes.

## 1. Introduction

Due to its geographical position and bioclimatic diversity, Morocco offers a rich, varied flora. It has 800 species of aromatic and medicinal plants alone [1], of which 100 species are exported as dried herbs [2]. Morocco, classified as the twelfth largest exporter of medicinal and aromatic plants in the world, still has considerable untapped potential. The Oriental region is home to the highest proportion of medicinal and aromatic plants exploited in the country, followed by the Fez-Meknes region [3].

*Rosmarinus officinalis* L. is popularly known as rosemary and, recently, a *Salvia rosmarinus* Spenn (*S. rosmarinus*), which belongs to the Lamiaceae family [4], grows best in hot climates, requires light and heat, and moderately tolerates drought. It is grown in withered, subhumid regions [2]. Rosemary is native to the Mediterranean regions from Spain to the Balkans and into North Africa [5]. In Morocco, rosemary is presented in the High and the Middle Atlas Mountains and the Oriental and the Rif regions [6]. However, it is rarely found in the western part of the country [1]. The Oriental region contains the largest amount of rosemary in the country [7]. Figuig Province holds 57% of rosemary resources, followed by Taourirt and Jerada [8]. Talsint commune produces more than 71% of the rosemary in Figuig [9]. 

The essential oil of rosemary is known by its chemical composition, which has beneficial properties. It is applied to cure several diseases, such as diseases related to inflammation [10], cancer [11], diabetes [12], cardiovascular diseases [13], and Alzheimer’s disease [14]. It is used for the treatment of respiratory and inflammatory diseases [15] due to the presence of the 1,8-cineole compound. Rosemary essential oil is also renowned for its antimicrobial activity [10,16], and this activity is associated with major chemical compounds of essential oil (1,8-cineole and α-pinene) [17]. The 1,8-cineole and α-pinene compounds are known for their antimicrobial activity against some microorganisms, such as *Bacillus subtilis* (Gram-positive), *Escherichia coli* (Gram-negative) [18], and *Mycobacterium smegmatis* (Gram-positive) [19]. Rosemary is used in the food industry as a preservative agent [20] and in cosmetics [4] as a stimulating and brightening agent [21], as well as a skin conditioning agent [22]. In Eastern Morocco, the production of rosemary generates about 81,000 days of work per year, with an equivalent value of $500,000 US. Through cooperatives located in Figuig, Taourirt, and Jerrada, the region has succeeded in building on its experience in this sector to enhance it into a source of richness for people by applying new cultivation methods for rosemary fields and adding new species from Asian countries [23]. 

The objective of this study was to compare the yield of 27 samples of rosemary essential oils and develop PCA and HCA methods to contribute to the knowledge of the effect of the harvest period and the cooperative on the yield and chemical quality of EOs from *S. rosmarinus*.

## 2. Results and Discussion

### 2.1. Variations of the Essential Oils Yield 

The yield obtained from the samples of rosemary ranged from 1.1 to 3.1% (Figure 1). The results showed that most of the periods had yields of higher than 2%, which is an encouraging factor for the future exploitation of this region. The yields of the three Moroccan cooperatives in this study were slightly higher than those reported by Sabbahi et al. [9], who found that the yield of rosemary essential oil in Talsint (located in the Figuig Province of Morocco) ranged from 0.6% to 1.7%. In Oujda, which is located in the Oriental region of Morocco, the yield of rosemary essential oil cultivated was about 1.8% [24]. Elyemni et al. [25] indicated that the rosemary yield of the region of Fez, Morocco, was 1.4%. Additionally, the yields of essential oils from four locations in Algeria were lower than those found in our study: 1.9% in the region of Ain Mlila commune (wilaya of Oum-El-Bouaghi), 1.6% in Bibans commune (wilaya of Bordj Bou Arreridj), 1.1% in Maadid commune (East wilaya of M’sila), and 0.7% in the area of Ain Turk commune (wilaya of Oran) [26]. In Tunisia, the yield of rosemary EOs ranged between 1.5% and 2.2% [27]. Statistical analysis revealed that the cooperative and the harvest period studied had a probability (*p*-value) of less than 5%. Consequently, they significantly affected the variable.

Figure 1 demonstrates the significant differences between the cooperatives and harvest periods in terms of the average yield of *S. rosmarinus* essential oil. There was a significant difference (*p* < 0.05) in the average yield of *S. rosmarinus* EO between cooperatives C1 and C2 as well as between cooperatives C1 and C3, while no significant differences (*p* > 0.05) were noticed among cooperatives C2 and C3 in terms of the average yield of *S. rosmarinus* EO. The highest average yields of the rosemary EOs was found in plants from cooperative C1. Regarding harvest periods, no significant differences were observed among periods P1 (January), P8 (February), P7 (December), and P6 (October) in terms of the average yield of *S. rosmarinus* EO. In addition, no significant differences were noted among periods P2 (March), P5 (September), P3 (May), and P9 (April), whereas significant differences were noted between the periods P2 (March), P4 (July), and P3 (May) in terms of the average yield of *S. rosmarinus* EO. However, highly significant differences were identified among P1 (January) and P9 (April), as well as between P8 (February) and P3 (May). The samples from Period 9 (collected in April) and the samples from Period 3 (collected in May) showed the highest average yields of *S. rosmarinus* EOs (3.1 ± 0.1 and 3.0 ± 0.1, respectively). The lowest average yield of *S. rosmarinus* EO (1.1 ± 0.1) was found in the plant collected in period 1 (January). According to two-way ANOVA, the yield of EOs was significantly affected by the harvest period and the cooperative. This finding may be explained by the environmental conditions in Figuig (a Saharan arid climate), especially the hot weather because the biosynthesis of EOs is more prominent in warm climates. This was shown by Rehman et al. [28], who found that the majority of enzymatic activities for the production of volatile compounds are enhanced by temperature. 

Accordingly, the yield of *S. rosmarinus* essential oil was maximized by using plants from cooperative C1 that were harvested in April. The variability of rosemary’s essential oil yields could have resulted from the origin of the plant, the harvest time [29,30], the environmental [27] and agronomic conditions [31,32], the plant’s phenological stage (Before and during full flowering stages) [33,34], and the extraction method [35].

### 2.2. Variations of the Essential Oil Compositions

Table 1 shows the results of the major components of the chemical composition of *S. rosmarinus* essential oils identified by gas chromatographic analysis coupled with mass spectrometry (GC/MS) for the three cooperatives during January 2018 and April 2019. Rosemary EOs contained different components according to the harvest period. Eight components were identified, representing a total of 83.7% for the three cooperatives. The major constituents were 1,8-cineole (28.6–51.1%), α-pinene (9.9–16.2%), camphor (5.3–16.8%), β-pinene (2.2–8.0%), camphene (2.3–7.7%), myrcene (0–4.5%), α-terpineol (0–3.8%), and limonene (0–3.3%) for all the samples during all periods. The essential oil from cooperative C1 showed the highest percentage of components. The 1,8-cineole component was present in a higher proportion than the other components, with a maximum value during period 3 (May, 2018); the α-pinene component reached its maximum value during period 6 (October 2018). 

The chemical compositions of the 27 individual *S. rosmarinus* EOs were identified as two chemotypes: 1,8-cineole and α-pinene. The 1,8-cineole chemotype was present in period 3 (May 2018), and the α-pinene chemotype was available in period 6 (October 2018). Our results were similar to those reported in [9] in Talsint in the Province of Figuig in the Oriental region of Morocco. In Israel, Sadeh et al. [36] identified 1,8-cineole, camphor, and α-pinene as the major compounds. In the Middle Atlas Mountains of Morocco, Hannour et al. [37] found that rosemary was characterized by the 1,8-cineole (46.2%), camphor (17.3%), borneol (6.8%), α-terpineol (5.3%), β-pinene (5.6%), camphene (2.6%), and terpinen-4-ol (2.2%) and that rosemary from Loukkos was characterized by camphor (21.3%), 1,8-cineole (17.0%), α-pinene (9.2%), β-pinene (8.6%), camphene (7.4%), terpinen-4-ol (2.8%), borneol (4.8%), and p-cymene (2.4%). Bouyahya et al. [38] showed that *S. rosmarinus* from Ouezzan Province in northern Morocco constituted a majority compound consisting of 1,8-cineole (23.7%), camphor (18.7%), borneol (15.5%), and α-pinene (14.1%). The main compounds of *S. rosmarinus* reported in [39] were camphor (31.2%), β-caryophyllene (18.6%), 2,4-hexadiene, 3,4-dimethyl-, (Z, Z) (9.1%), α-fenchene (4.7%), cis-verbenone (4.3%), and bornyl acetate (3.4%). In Tunisia, the major components indicated in [40] were α-pinene (12.6–42.8%), α-fenchene (1.2–2.2%), 1,8-cineole (20.8–64.7%), camphor (14.5–20.4%), isoborneol (2.3–9.8%), and myrtenal (4.3–7.4%). Hendel et al. [26] identified camphor (35.3–37.6%), camphene (18.1–22.4%), α-pinene (16.1–21.0%), 1,8-cineole (12.1–14.4%), limonene (2.3–4.3%), ρ-cymene (0.5–2.6%), α-terpineol (0.7–1.2%), and borneol (0.8–3.2%) as the major compounds of Algerian rosemary essential oil. Oils of *S. rosmarinus* from Spain were characterized by the presence of camphor (17.2–34.7%), α-pinene (15.8–21.6%), 1,8-cineole (12.1–14.4%), camphene (5.2–8.6%), borneol (3.2–7.7%), β-pinene (2.3–7.5%), verbenone (2.2–5.8%), β-myrcene (0.9–4.5%), limonene (2.0–3.8%), bornyl acetate (0.2–2.3%), α-terpineol (1.2–2.5%), and ρ-cymene (0.2–1.7%) [41]. In Italy, Leporini et al. [42] mentioned 1,8-cineole (21.89–16.98%), camphor (11.08–7.27%), trans-caryophyllene (10.58–8.62%), α-pinene (10.96–10.37%), camphene (6.87–6.30%), borneol (3.31–5.30%), α-terpineol (3.19–4.05%), sabinene (1.01–2.82%), myrcene (1.32–2.73%), thujene (0.88–2.34%), γ-terpinene (2.42–2.76%), limonene (1.78–2.30%), and α-terpinene (1.19–1.35%) as the compounds of rosemary EOs. Pitarokili et al. [43] found 1,8-cineole (48.3–58.7%), borneol (8.8–10.4%), α-Pinene (8.4–9.9%), α-terpineol (4.3–5.9%), camphene (2.2–3.5%), β -caryophyllene (0.7–4.4%), bornyl acetate (0.7–3.4%), and ρ-cymene (1.7–3.1%) as the major compounds of rosemary Eos from Greece, whereas, ρ-cymene (44.02%), linalool (20.5%), terpinene (16.62%), β-pinene (3.61%), α-pinene (2.83%), eucalyptol (2.64%), and thymol (1.81%) were the most common compounds of Turkish rosemary EOs [44]. The quantity of components varied according to the isolation method used. The chemical composition of *S. rosmarinus* is highly sensitive. The literature reports that several factors can modify its quality and quantity: the method of extraction [36], the harvest period [45], the environmental conditions [29], the site of collection [30], the harvest stage [29,46], and the plant’s genotype [33].

**Table 1 molecules-27-02914-t001:** Major components of the chemical composition of *S. rosmarinus* essential oils from each cooperative and harvest period.

	Compounds	RI *	RI Lit *	Cooperative	Harvest Period								
					Period 1	Period 2	Period 3	Period 4	Period 5	Period 6	Period 7	Period 8	Period 9
**% Relative Peak Area**	α-pinene	939	938	C1 *	14.8 ± 3.5	12.1 ± 1.2	9.9 ± 0.2	11.9± 0.2	12.4 ± 0.7	15.4 ± 1.4	15.0 ± 1.1	15.0 ± 1.1	12.9 ± 0.1
				C2	14.1 ± 3.4	12.1 ± 0.0	11.6 ± 0.5	13.7± 0.5	13.8 ± 0.5	14.8 ± 1.6	14.6 ± 1.5	14.6 ± 1.5	13.1 ± 1.0
				C3	13.5 ± 2.1	11.9 ± 0.5	10.7 ± 0.5	11.7 ± 0.3	13.2 ± 0.3	16.2 ± 0.3	14.5 ± 0.4	14.6 ± 0.4	12.7 ± 0.4
	Camphene	953	952	C1	3.5 ± 1.1	3.5 ± 1.1	2.9 ± 0.7	2.9 ± 0.4	3.9 ± 1.4	5.0 ± 0.6	3.4 ± 0.0	4.6 ± 0.6	4.1 ± 1.5
				C2	6.8 ± 0.3	6.3 ± 0.2	6.3 ± 0.2	4.8 ± 0.7	3.8 ± 0.3	7.2 ± 0.5	4.2 ± 0.6	5.3 ± 1.3	5.1 ± 1.0
				C3	7.4 ± 0.1	5.9 ± 0.4	5.9 ± 0.4	4.9 ± 0.8	4.7 ± 0.1	6.5 ± 1.0	4.6 ± 1.1	4.6 ± 0.5	5.4 ± 0.1
	β-pinene	976	980	C1	7.1 ± 0.1	6.1 ± 1.0	6.1 ± 1.0	6.1 ± 1.0	6.8 ± 0.7	4.4 ± 1.1	4.3 ± 0.9	6.8 ± 0.6	6.1 ± 0.1
				C2	7.3 ± 0.6	7.3 ± 0.6	7.3 ± 0.6	7.3 ± 0.6	6.1 ± 0.3	4.1 ± 1.8	3.6 ± 1.5	5.1 ± 0.9	6.7 ± 0.2
				C3	6.9 ± 1.2	6.9 ± 1.2	6.8 ± 1.2	6.8 ± 1.2	6.5 ± 1.2	6.5 ± 1.2	6.1 ± 1.1	6.7 ± 0.4	6.5 ± 0.6
	Myrcene	990	993	C1	3.0 ± 2.6	3.0 ± 2.6	2.9 ± 1.7	1.9 ± 0.6	1.6 ± 0.1	1.6 ± 0.1	2.6 ± 0.1	1.1 ± 0.1	2.7 ± 1.8
				C2	1.2 ± 1.0	0.0 ± 0.0	1.0 ± 0.1	1.6 ± 0.4	1.6 ± 0.1	1.6 ± 0.1	2.6 ± 0.1	1.1 ± 0.1	1.2 ± 0.4
				C3	1.3 ± 1.1	1.3 ± 1.1	1.4 ± 0.5	1.7 ± 0.5	1.9 ± 0.5	1.9 ± 0.4	2.4 ± 0.0	1.3 ± 0.2	1.5 ± 0.3
	Limonene	1030	1031	C1	0.0 ± 0.0	1.9 ± 1.8	1.9 ± 0.1	2.5 ± 0.4	1.5 ± 0.4	1.5 ± 0.4	1.4 ± 0.4	0.9 ± 0.0	2.8 ± 0.5
				C2	0.1 ± 0.1	1.0 ± 0.8	1.6 ± 0.4	1.4 ± 0.1	1.3 ± 0.2	1.3 ± 0.2	1.4 ± 0.1	0.8 ± 0.1	1.5 ± 0.7
				C3	0.0 ± 0.0	1.1 ± 0.7	1.8 ± 0.0	2.4 ± 0.5	1.9 ± 0.1	1.9 ± 0.1	1.5 ± 0.0	0.7 ± 0.1	1.0 ± 0.1
	1,8-cineole	1033	1033	C1	33.2 ± 0.7	39.2 ± 0.3	51.1 ± 1.8	47.1± 2.8	42.3 ± 1.0	37.8 ± 0.5	32.3 ± 1.0	33.2 ± 0.7	43.4 ± 1.1
				C2	29.8 ± 1.8	37.8 ± 0.3	46.2 ± 0.3	42.2 ± 0.3	41.9 ± 0.6	35.9 ± 0.4	31.8 ± 0.6	29.8 ± 1.8	41.1 ± 2.0
				C3	28.6 ± 0.8	38.6 ± 0.8	47.3 ± 0.8	43.1 ± 1.3	42.2 ± 0.4	34.2 ± 0.7	32.1 ± 0.3	28.6 ± 0.8	38.5 ± 0.6
	Camphor	1143	1144	C1	14.3 ± 0.3	10.8 ± 1.1	7.3 ± 0.5	8.3 ± 0.5	7.2 ± 0.6	16.8 ± 2.3	18.2 ± 1.8	18.8 ± 0.3	5.9 ± 1.2
				C2	12.8 ± 1.2	12.8 ± 1.2	5.8 ± 0.2	6.8 ± 0.2	6.3 ± 0.3	16.8 ± 0.8	17.8 ± 0.9	18.8 ± 0.1	8.7 ± 0.7
				C3	15.5 ± 0.0	15.3 ± 0.1	5.3 ± 0.1	8.8 ± 2.4	7.8 ± 1.2	15.4 ± 0.1	16.4 ± 1.1	17.4 ± 1.1	10.0 ± 0.5
	α-Terpineol	1185	1189	C1	2.2 ± 1.6	1.7 ± 1.1	1.4 ± 0.2	1.2 ± 0.2	1.2 ± 0.1	0.8 ± 0.2	0.8 ± 0.1	0.8 ± 0.1	1.6 ± 0.5
				C2	1.7 ± 0.1	1.7 ± 0.1	1.2 ± 0.0	1.4 ± 0.1	1.3 ± 0.1	0.9 ± 0.1	0.9 ± 0.1	0.9 ± 0.1	1.1 ± 0.1
				C3	0.0 ± 0.0	1.4 ± 0.5	1.1 ± 0.1	1.3 ± 0.1	1.3 ± 0.0	0.9 ± 0.0	0.9 ± 0.0	0.9 ± 0.0	0.4 ± 0.4
	Total%			C1	78.1 ± 2.3	78.4 ± 2.3	83.3 ± 0.6	81.9 ± 1.8	77.0 ± 1.9	83.4 ± 0.6	78.2 ± 0.3	81.2 ± 2.6	79.5 ± 0.2
				C2	73.8 ± 4.6	78.9 ± 0.0	81.0 ± 0.7	79.2 ± 0.5	76.2 ± 1.4	82.5 ± 2.3	76.9 ± 0.9	76.1 ± 2.3	78.6 ± 2.5
				C3	73.2 ± 0.1	82.2 ± 3.3	80.3 ± 0.9	80.7 ± 4.0	79.4 ± 1.5	84.0 ± 2.1	78.4 ± 3.1	74.7 ± 1.1	76.0 ± 0.2

Data expressed as mean ± standard deviation of triplicates. * Cn: cooperative. * RI: retention indices calculated experimentally using homologous series of C8-C28 alkanes. * RI Lit: retention indices from the literature [47]. Compounds and their percentage value determined from the chromatograms of three experimentations obtained on an HP-5 MS column.

### 2.3. Principal Component Analysis

Principal component analysis (PCA) allowed us to identify the correlation among the major compounds and the yield of rosemary essential oil from different cooperatives and harvest periods, as well as to classify similar samples in accordance with the yield and the major compounds of rosemary essential oil. PCA also enabled the explicit representation of individuals and variables to highlight the different profiles between samples according to the harvesting period and the cooperative. The individuals are represented by the 27 essential oil samples taken from three cooperatives during nine different harvest periods. The variables are represented by the average yield in essential oil and the main components of the chemical composition of rosemary. To verify the feasibility of the PCA, the Kaiser–Meyer–Olkin (KMO) index and the Bartlett test were used. Calculation of the KMO index resulted in a value of 0.8, which is greater than 0.5; and Bartlett test has obtained a probability of less than 5%, meaning the test is highly significant. Therefore, the PCA is attainable. 

To determine the number of principal components, the components with eigenvalues were greater than one had to be retained [48]. Table 2 demonstrates that the first three principal components had eigenvalues greater than one and that they explained most of the variables. The cumulative percentage of these three components explained 82.9% of the data variability. The first component explained 47.5% of the variables: α-pinene, 1,8-cineole, camphor, α-terpineol, and yield. The second component explained 22.5% of the variables: camphor, β-pinene, and limonene. The third component explained 12.9% of the variables: myrcene.

The loading plot or distribution of the variables according to the first principal component (PC1) and the second principal component (PC2) (Figure 2) shows that the variables α-pinene, 1,8-cineole, camphor, yield, and α-terpineol are well represented in axis 1 (PC1), and the variables camphor, β-pinene, and limonene are well represented in axis 2 (PC2). Thus, the loading plot also shows the presence of some correlation among the studied variables. A positive correlation was identified among the yield of essential oil, α-terpineol, and 1,8-cineole. α-pinene and camphor also showed a positive correlation. However, 1,8-cineole and camphor were negatively correlated. Another negative correlation was observed between α-pinene and the yield of essential oil. These results indicate that when the yield of essential oil increased, the amount of 1,8-cineole increased. However, when the quantity of 1,8-cineole contained in essential oil was higher, the quantity of α-pinene and camphor was lower.

The score plot (Figure 3) or the distribution of individuals (harvest period and cooperative) in accordance with PC1 and PC2 shows that individuals were separated into two main groups. Group A consists of P9 (April), P3 (May), P4 (July), and P5 (September), or the summer season. Group B is composed of P6 (October), P1 (January), P7 (December), and P8 (February), or the winter season. The intermediary between the two groups was P2 (March), which shared both of their characteristics. No significant difference was found between the essential oils of the three cooperatives because the distribution of cooperatives in the groups was random. This finding may also be due to the locations of the cooperatives being in the same province (Figuig).

To obtain the biplot, the individual factors were projected on the variables (Figure 4). The graph shows that group A is dominated by the highest amount of 1,8-cineole and α-terpineol compounds and a high yield of essential oil. However, group B presented the opposite trend of group A, which is characterized by α-pinene and camphor compounds, whereas group B was characterized by a low yield of essential oil and a low amount of 1,8-cineole and α-terpineol compounds, especially in P9 (October) and P7 (December) individuals. Period 2 (March) was characterized by an average yield of essential oil and an average amount of the constituents of both groups. Several studies have shown the effect of the harvest period and climatic conditions on the level of certain compounds in rosemary essential oil [34]. Melito et al. [49] found that summer had the highest yield of *S. rosmarinus* essential oils, whereas winter had the lowest yield. These authors confirmed that the chemical composition of *S. rosmarinus* was significantly influenced by the seasons, observing that γ-terpinene and terpinolene values were high in summer (0.8 and 0.7, respectively) and lowest in winter. They also reported that β-pinene and bornyl acetate components were the most elevated in the spring. Ismaili et al. [50] reported that the best harvest time was spring, with a yield of 2.0%, and they also showed that the chemical composition did not vary according to the season. Lemos et al. [51] noted that yields varied during the harvest period, with the highest yield of rosemary essential oil occurring in April (0.90%). The authors of [34,51] showed that rainfall and light intensity directly influence the production of essential oil.

### 2.4. Hierarchical Cluster Analysis

To better visualize the classification of the studied samples (harvest period) according to the main components of chemical composition and the yield of rosemary essential oil, an HCA was performed (Figure 5). As a validation of PCA results (scores plot), the individuals were divided into two main groups: Group A represented summer (periods P2, P3, P4, P5, and P9) and Group B represented winter (periods P2, P6, P7, and P8). March (P2) was included in both groups. Thus, it represented the intermediary between the two groups. Group A (winter season) was characterized by a high amount of α-pinene and camphor, a low amount of 1,8-cineole and α-terpineol compounds, and a low yield of essential oil. However, group B (summer season) was characterized by a high amount of 1,8-cineole and α-terpineol compounds, a high yield of essential oil (red color), and a low amount of α-pinene and camphor (blue color). The intermediate group (P2: March) was characterized by an average yield of essential oil and an average amount of *S. rosmarinus* main compounds. The result of both PCA and HCA indicated that the yield and chemical composition of essential oil were significantly influenced by the harvest period and environmental conditions. The majority of studies indicate that the yield of essential oil and the chemical composition of *S. rosmarinus* are impacted by environmental conditions. Bajalan et al. [52] found a correlation between the major compounds of rosemary essential oil and environmental factors. Sarmoum et al. [53] indicated that the production of rosemary essential oil was directly affected by water stress; this explains why the yield of rosemary essential oil was higher in the summer season and lower in the winter season. Nevertheless, Sadeh et al. [36] showed that the main compounds of rosemary essential oil were not significantly affected by season. 

### 2.5. Antimicrobial Activity

The essential oils tested were composed of the average of the samples collected during the third period (May 2018), which belongs to summer season (high yield of *S. rosmarinus* essential oil and the highest proportion of 1,8-cienole compound). The antimicrobial activity was first qualitatively tested by the disc diffusion technique with the aim of selecting the most active EOs among those tested. These samples were evaluated against one Gram-negative (*Escherichia coli*) and two Gram-positive (*Bacillus subtilis*) (*Mycobacterium smegmatis*) species.

Figure 6 illustrates the significant differences (*p* < 0.05) in the diameter of inhibition zones of the rosemary essential oil from three cooperatives: C1, C2, and C3. Significant differences were observed among the essential oil from all cooperatives and antibiotics in terms of all bacteria strain inhibition zones. However, no significant difference (*p* > 0.05) was noted between the essential oil from cooperatives C1 and C3, whereas a significant difference was found among the essential oils from the cooperatives C1 and C2, as well as antibiotics in terms of *E. coli* inhibition zones. In terms of inhibition zones of *B. subtilis*, a significant difference was noticed between essential oil from the cooperatives C1, C2, and C3, whereas no significant difference was observed between essential oils from the cooperatives C2 and antibiotics. Regarding *M. smegmatis*, again, significant differences were identified among the essential oils from the cooperatives C1 and C2, as well as antibiotics, whereas no significant difference was identified between the essential oils from cooperatives C2 and C3. Two-way ANOVA showed that there was a statistically significant difference among the means of the inhibition diameters and three cooperative EOs for the three strains used.

Figure 6 shows that the essential oil from cooperative C1 had the highest activity, with inhibition zones of 16.6 ± 0.5 mm, 31.3 ± 2.1 mm, and 37.1 ± 1.5 mm for *E. coli* ATCC25922, *B. subtilis* ATCC 23857, and *M. smegmatis* MC2 155, respectively. In the essential oil from cooperative C2, the greatest activity was noted against *M. smegmatis*, with an inhibition diameter of 26.2 ± 1.1 mm. The lowest activity was observed against *E. coli*, with inhibition diameters of 19.0 ± 1.0 mm. The same results were observed in the essential oil from cooperative C3. The inhibition diameters obtained for M. *smegmatis* (27.0 ± 1.1 mm) were higher than those obtained for *B. subtilis* (18.6 ± 2.1 mm) and *E. coli* (15.2 ± 1.1 mm). Many studies have demonstrated that rosemary essential oil provides effective antibacterial activity against various microorganisms [38,40]. Rosemary essential oil was slightly less sensitive against Gram-negative bacteria (*E. coli*) than they against Gram-positive bacteria [54]. Megzari et al. [55] observed that rosemary essential oil presented antibacterial activity against the same bacterial strain, with diameters of inhibition similar to those observed in our study. However, Bajalan et al. [56] stated that the essential oil of rosemary showed antibacterial activity against the four bacteria strains, especially *E. coli*, with an inhibition zone of 18.5 mm, followed by *Staphylococcus aureus* (14.6 mm), *Klebsiella pneumoniae* (13.9 mm), and *Streptococcus agalactiae* (13.1 mm). Abdellaoui et al. [57] revealed that the highest antibacterial activity was against *E. coli*, which was the most sensitive of the tested strains with the largest inhibition zone (18.4 mm). Chahboun et al. [58] showed that rosemary essential oil in the Taza region of Morocco was active against all strains, except for *Pseudomonas aeruginosa*, with the other strains having diameters between 12.0 and 22.0 mm. 

The results of the minimum inhibition concentration are represented in Table 3. The essential oils showed important inhibitor activity against the microorganisms studied. In fact, the bacterial strains as a whole were inhibited at a 1/250 *v*/*v* concentration. *M. smegmatis* and *B. subtilis* were the most susceptible strain for all Eos, with growth ending at 1/500 (*v*/*v*). *E. coli* was the least sensitive microorganism, an MIC value of with 1/250 (*v*/*v*). All EOs exerted strong activity against *M. smegmatis*. Fadil et al. [59] reported that rosemary essential oil exhibited antimicrobial activity against *Salmonella typhimurium*, with an MIC value of 2% (*v*/*v*).The MIC value of rosemary essential oil reported in [60] against three bacterial strains (*S.aureus*, *E.coli*, and *Pseudomonas aeruginosa*) was 3.33, 1.67, and 6.67 µL/mL, respectively. According to [61], rosemary essential oil showed antimicrobial activity *against E. coli, Salmonella choleraesuis, Listeria monocytogenes*, and *S. aureus*, with an MIC value of 0.65, 0.65, 0.91, and 0.91 mg/mL, respectively. 

It can be concluded that the rosemary essential oil from cooperative C1 exhibited the best antimicrobial activity of the three cooperatives. This can be explained by a high amount of 1,8-cineole compound in rosemary essential oil from cooperative C1 (51.1 ± 1.8%) compared to the cooperatives C2 and C3 (46.2 ± 0.3 and 47.3 ± 0.8%, respectively), but the contributions of α–pinene and camphor compounds should also be noted. In addition, the analysis of the *S. rosmarinus* essential oil from cooperative C1 showed a high rate of camphor (7.3 ± 0.5%) compared to cooperatives C2 (5.8 ± 0.2%) and C3 (5.3 ± 0.1%). The essential oil from cooperative C2 represented the highest amount of α-pinene compound (11.6 ± 0.5%), followed by C3 (10.7 ± 0.5%) and C1, with a value of 9.9 ± 0.2%. The effect of synergy could also be the origin of this activity. Several authors demonstrated that chemical composition has an effect on antimicrobial activity, such as 1,8-cineole and α-pinene [62,63]. According to [64], the antibacterial activity against *M. smegmatis* of rosemary was most likely to occur when high quantities of 1,8-cineole were present. Moreover, Kovač et al. [65] demonstrated that the α-pinene compound exhibited antimicrobial activity against 16 strains of *Campylobacter jejuni*. Ojeda-Sana et al. [66] indicated that rosemary essential oil has a high antibacterial activity against Gram-positive bacteria (*S. aureus* and *Enterococcus faecalis*) and against the Gram-negative bacteria (*E. coli* and *K. pneumoniae*). The same authors also demonstrated that the α-pinene compound presented a broad antibacterial spectrum against Gram-negative and Gram-positive bacteria and that the 1,8-cineole compound showed antibacterial activity against pathogenic bacteria by causing a rupture of the cell membrane. It is known by the scientific community that the main compounds of EOs have a key role in antibacterial activity. However, the importance of minor components or a combination thereof could explain the complete activity of EOs against clinical and foodborne pathogens [64,67]. 

The present study indicates that the most important period to exploit the essential oil of *S. rosmarinus* is in summer for the three studied cooperatives in Figuig Province. This is due to the abundance of the main components that affect antibacterial activity, including 1,8-cineole, α-pinene, and camphor. 

## 3. Materials and Methods

### 3.1. Plant Material

The aerial parts of *Salvia rosmarinus* Spenn (*S. rosmarinus*) were collected from three cooperatives (Table 4) located in the Figuig Province in the Oriental region of Morocco (Figure 7). *S. rosmarinus* plants were collected from wild populations during nine periods from January 2018 to April 2019 (P1: January 2018, P2: March 2018, P3: May 2018, P4: July 2018, P5: September 2018, P6: October 2018, P7: December 2018, P8: February 2019, P9: April 2019) on the same day from the three cooperatives. Botanical identification was carried out by Professor Ghanmi Mohammed (a former researcher at the Forestry Research Center, Rabat, Morocco). The variables studied were the average percentage yield and the chemical composition of essential oils of the plant for each harvest period and cooperative.

### 3.2. Essential Oil Extraction

The aerial parts of the plants were dried in shade at ambient temperatures and then hydro distilled using a Clevenger-type apparatus [69]. This method is recommended for essential oil extraction because it is simpler and faster than other extraction methods, and it allows for a good yield and recovery of the compounds of the essential oil [70]. Each specimen was put in a 2 L glass flask containing 100 g of dry plant material and 800 mL of distilled water [71]. The mixture was then brought to a boil for three hours. The essential oil obtained was placed in an opaque green flask, dried using anhydrous sodium sulfate, and stored at 4 °C [72]. The yields of the EOs from the different samples were calculated using formula (1) indicated by [73]. All the experiments were carried out in triplicate.
(1)$$Yield\%=\frac{weight\;of\;EO\;obtained\;by\;distillation\;\left(g\right)}{weight\;of\;dry\;biomass\;\left(g\right)}\times 100$$

### 3.3. Gas Chromatography/Mass Spectrometry Analysis 

Gas chromatography analyses for all the samples were undertaken on a Hewlett-Packard (HP 6890) gas chromatograph equipped with an HP-5 capillary column (30 m × 0.25 mm, film thickness of 0.25 µm), an FID detector, and an injector fixed at 275 °C. The oven temperature was set at 50 °C for five min and was then raised to 250 °C at a rate of 4 °C/min. The carrier gas applied was N2 at 1.8 mL/min, employing a split mode of ratio: 1/50, flow: 72.1 mL/min. The samples were diluted to 1/50 in methanol, and the injection was added manually with an injected volume of 1.2 µL.

The chemical composition was realized by gas chromatography coupled with mass spectrometry (GC/MS). This analysis was performed on a Hewlett-Packard gas chromatograph (HP 6890) coupled with a mass spectrometer (HP 5973). The column used was a capillary column equipped with an HP-5MS (cross-linked 5% PHME siloxane) (30 m × 0.25 mm, film thickness 0.25 µm). The column temperature was set at 50 °C and raised to 250 °C at a rate of 2 °C/min. The carrier gas was He at 1.5 mL/min, and a split mode ratio was employed: 1/74.7, flow: 112 mL/min. The MS identity of the components was confirmed using the NIST 98 spectra library. The parameters exploited in the mass spectra were ionization voltages of 70 eV, the ion source temperature was 230 °C, and the scan mass range was 35–450 m/z. The identification of the components was also verified by the comparison of the compounds’ elution order with their relative retention indices indicated in the literature. Each experiment was performed in triplicate.

### 3.4. Antimicrobial Activity

#### 3.4.1. Bacterial Strains

The antibacterial activities of rosemary essential oils from three cooperatives in Figuig Province in eastern Morocco were evaluated against three bacterial strains: *Escherichia coli* ATCC25922 (Gram-negative), *Bacillus subtilis* ATCC 23857 (Gram-positive), and *Mycobacterium smegmatis* MC2 155 (Gram-positive). These strains were taken from the culture collections of the Laboratory of Microbial Biotechnology at the Faculty of Sciences and Techniques in Fez, Morocco. 

#### 3.4.2. Disc Diffusion Method

The antimicrobial activity for each essential oil was determined using a disc diffusion method based on that used in [74] to identify the antibacterial power of the oils. The bacterial suspension was adjusted to 10^8^ CFU/mL and was then spread on a plate containing Muller–Hinton agar (MHA). Whatman paper sterile discs (6 mm in diameter) were soaked with 10 µL of essential oil samples and put on the inoculated MHA. In addition, the negative control was prepared in the same manner as the experimental test, and sterile distilled water was added rather than essential oil. A commercial disc of amoxicillin (30 μg and 86.2% purity) was used as a positive reference standard to determine the sensitivity of the tested strain. The diameters of the inhibition zone around the disks were measured after 24 h incubation at 37 °C. All assays were realized in triplicate.

#### 3.4.3. Minimal Inhibitory Concentration

The minimal inhibitory concentration (MIC) of essential oils was determined in a 96-well microplate using the microdilution method, according to [75], with a slight modification. The EOs were diluted sequentially in Mueller–Hinton broth (MHB) with agar added at 0.15% *w*/*v* as an emulsifier. The final concentration of the essential oil was between 1/10 *v*/*v* and 1/2000 *v*/*v*. The 12th was regarded as a growth control, as it only contained the culture medium and the strain. Afterwards, 50 mL of bacterial suspension was supplemented in each well at a final concentration of 10^6^ CFU/mL. After incubation at 37 °C for 18 h, the MIC was presented as the lowest essential oil concentration that showed a negative bacterial growth translated by a non-change in resazurin color. A positive MIC is detected by a reduction of blue dye resazurin to pink resorufin. Each experiment was performed in triplicate.

### 3.5. Statistical Analysis

To examine the significant differences between mean values, an analysis of variance (two-way ANOVA), and a Tukey’s post hoc test were utilized (a probability of *p* ≤ 0.05 was considered statistically significant) using Origin Pro 2021 software. The percentage composition of 8 major compounds (1,8-cineole, camphor, α-pinene, camphene, β-pinene, myrcene, α-terpineol, and limonene) and the average yield of rosemary essential oil were calculated to determine the variable variation and the correlation between them, as well as to identify the similarities between samples (harvest period and cooperatives) with principal component analysis (PCA). For better visualization of the similarity between harvest periods, a hierarchical cluster analysis (HCA) was performed. This classification tool divided the groups according to the average yield and the 8 major components of rosemary essential oil. The PCA and the HCA were realized using SPSS (version 25) and JMP Pro 14 software.

## 4. Conclusions

In this study, we focused on explaining the variations in rosemary essential oil yield, chemical composition, and antibacterial activity in relation to cooperatives and harvest periods. The best yield was obtained in essential oils for plants from cooperative C1 for period 9 (April 2019). The highest amount of the 1,8-cineole component appeared in period 3 (May 2018). PCA and HCA indicated the presence of two groups: group A (winter), which was characterized by camphor and α-pinene, and group B (summer), which was characterized by a high quantity of 1,8-cineole and high yield of rosemary essential oil. These statistical analyses indicated that summer is the best harvest period to exploit the essential oil of this plant in Figuig Province. The antibacterial activity showed an inhibitory potential against all strains, especially among samples from the cooperative C1. Our findings could have a positive economic impact on the exploitation of rosemary in the Oriental region, especially during the best harvest periods, as they show the best yields and the oils richest in 1,8-cineole and α-pinene chemotypes. The use of chemometric tools has the potential to identify the compounds responsible for antibacterial activities and will also make it possible to exploit these results to determine the most active oil fractions against pathogenic strains. In this context, this study could be extended to other bacterial strains of medical and food interest.

## Figures and Tables

**Figure 1 molecules-27-02914-f001:**
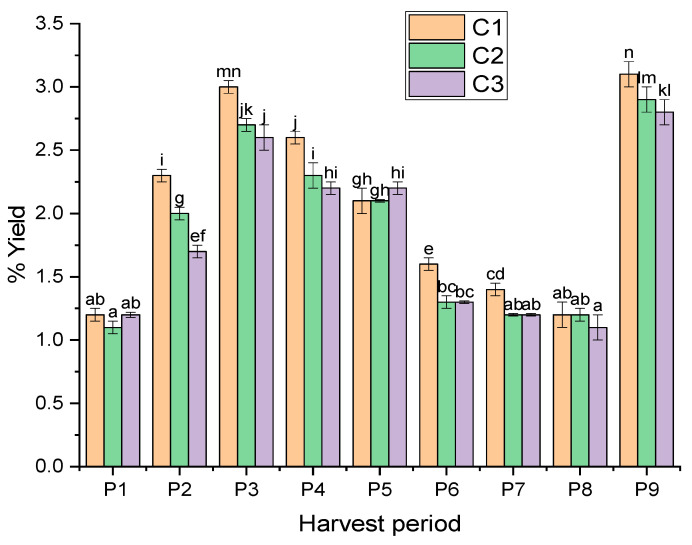
Average yield according to the essential oils from the three cooperatives during nine harvest periods. Each column represented by different letters (a, b, c, d, e, f, g, h, i, j, k, l, m, and n) indicates a significant difference (*p* < 0.05) based on Tukey test. Cn: cooperatives, Pn: periods.

**Figure 2 molecules-27-02914-f002:**
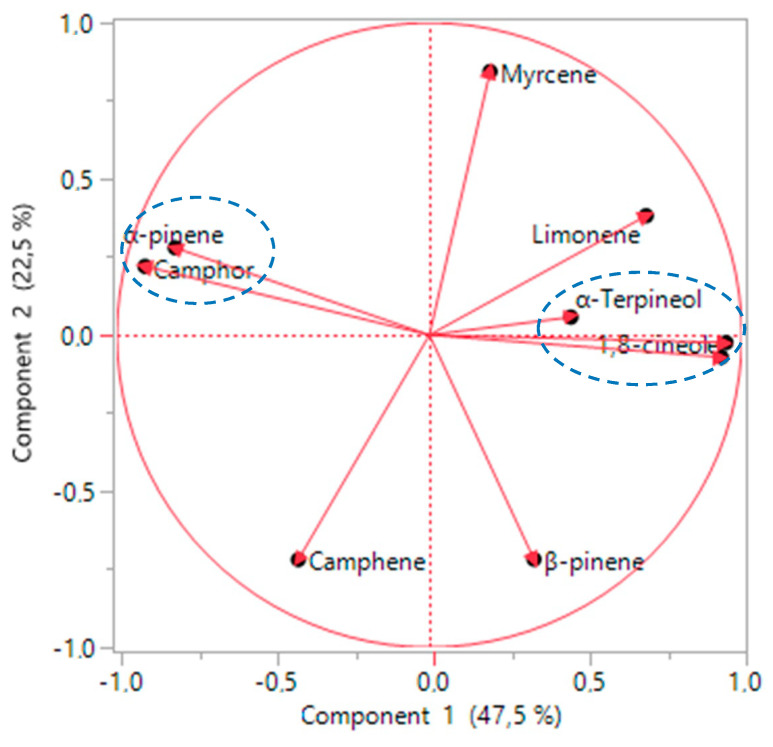
Loading plot that presents the cloud of variables in the first and second principal components.

**Figure 3 molecules-27-02914-f003:**
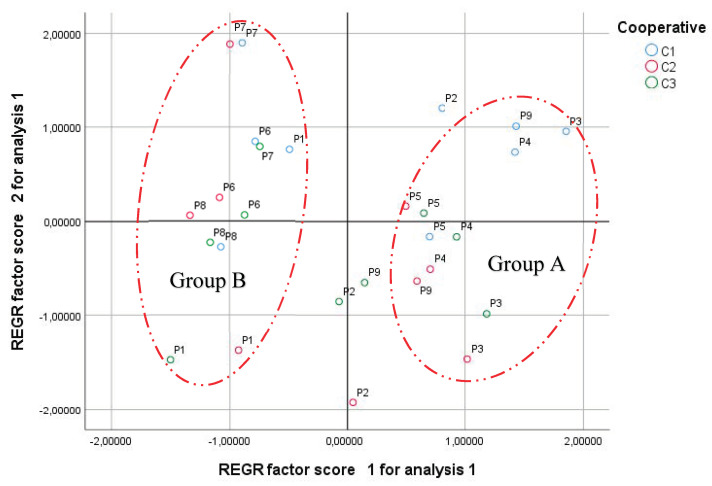
Projection of the cloud of individuals (score plot) in the first and second principal components. Pn: period of harvesting. Group A represents the summer season. Group B represents the winter season.

**Figure 4 molecules-27-02914-f004:**
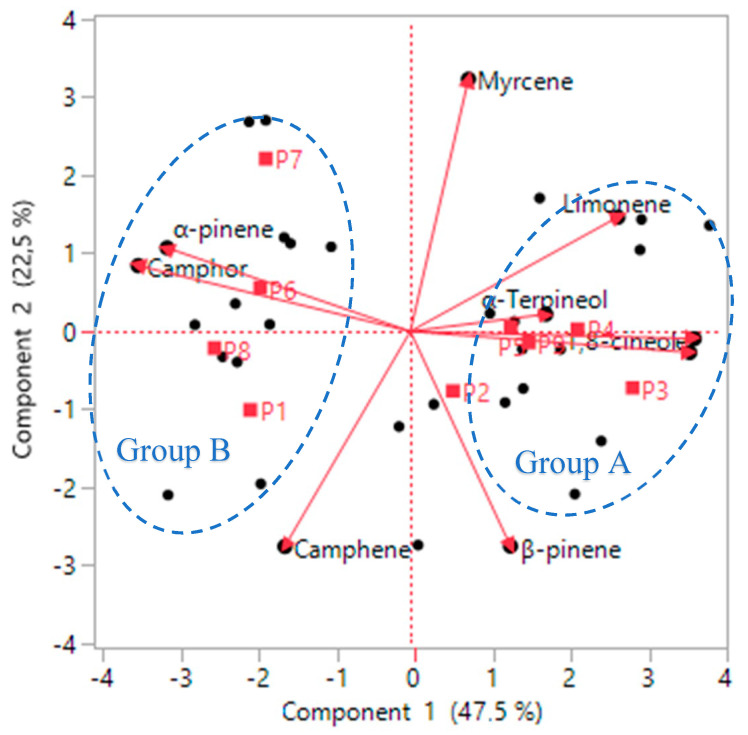
Biplot of variables and individuals in the factorial design. Pn: period of harvesting. Group A represents the summer season. Group B represents the winter season.

**Figure 5 molecules-27-02914-f005:**
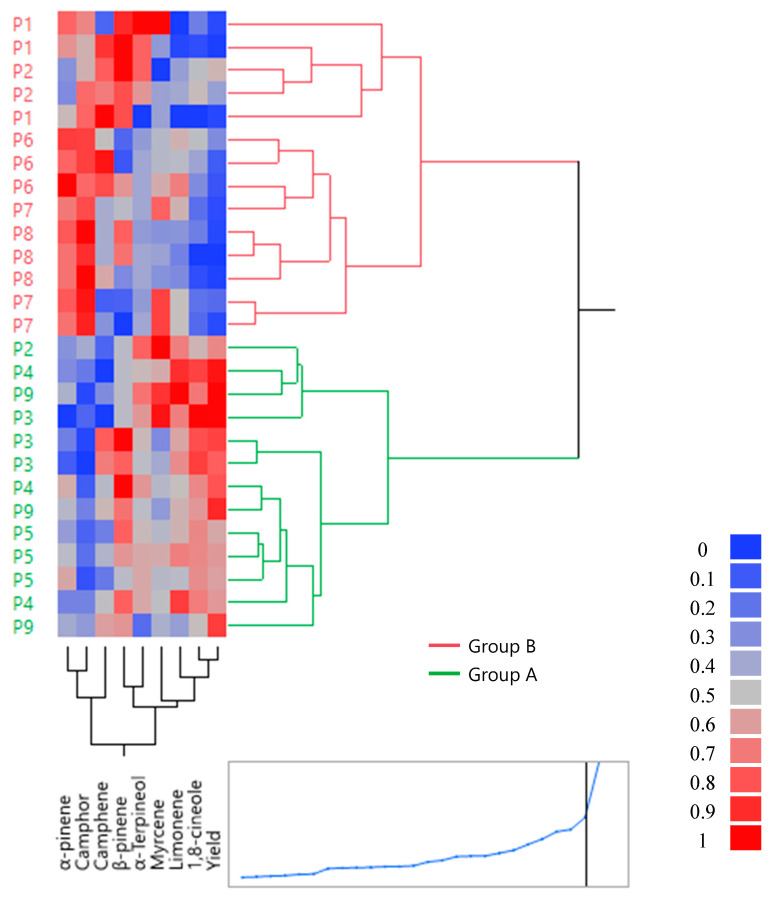
Hierarchical group analysis of samples of *S. rosmarinus* based on the method of ward. Group A represents the summer season. Group B represents the winter season.

**Figure 6 molecules-27-02914-f006:**
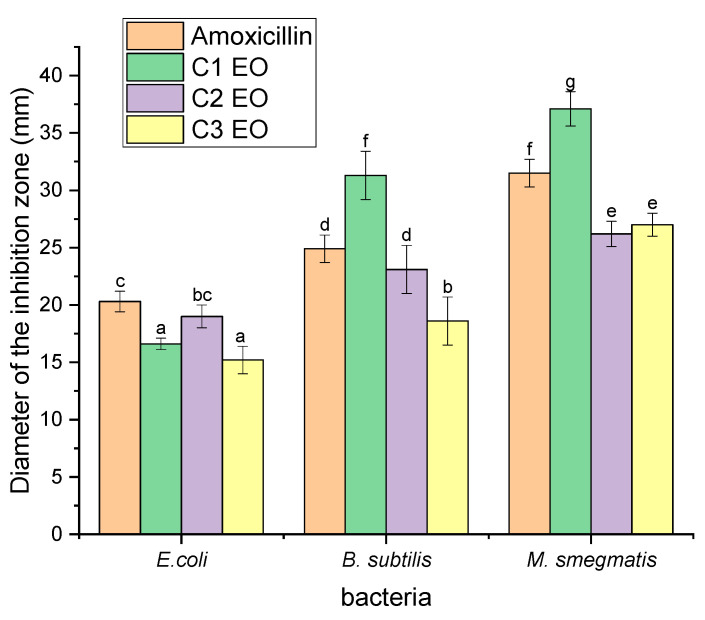
Diameter of inhibition zones of three bacteria strains as a function of the essential oils from the three cooperatives. Each column represented by different letters (a, b, c, d, e, f, and g) represent a significant difference (*p* < 0.05) based on Tukey test. Cn EO: essential oil from different cooperatives.

**Figure 7 molecules-27-02914-f007:**
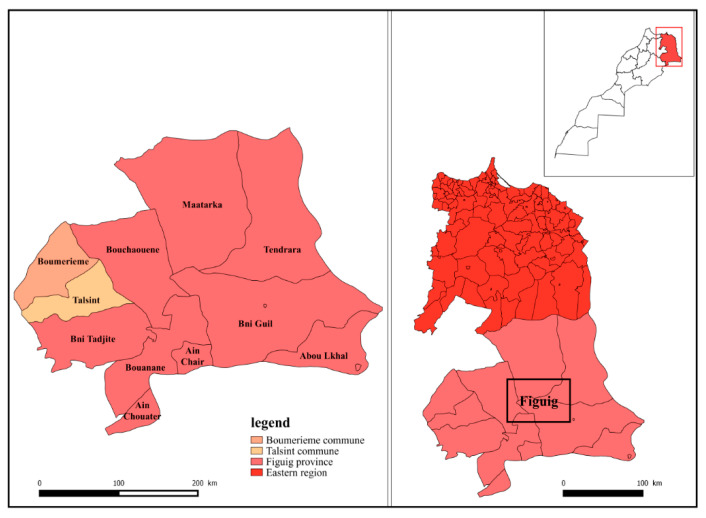
Geographical localizations of the study area: Figuig Province, eastern Morocco [68]. Made with QGIS software.

**Table 2 molecules-27-02914-t002:** Quality of representation of the variables on the first three components, eigenvalues of the components, percentages of variability explained, and cumulative percentages.

	Components		
	PC1 *	PC2	PC3
α-pinene	−0.8	0.3	0.1
Camphene	−0.4	−0.7	−0.3
β-pinene	0.3	−0.7	0.5
Myrcene	0.2	0.8	0.2
Limonene	0.7	0.4	−0.4
1,8-cineole	0.9	−0.0	−0.2
Camphor	−0.9	0.2	−0.0
α-Terpineol	0.5	0.1	0.8
Yield	0.9	−0.1	−0.2
Eigenvalue	4.3	2.0	1.2
Percentage of explained variability (%)	47.5	22.5	12.9
Cumulative percentage (%)	47.5	70.0	82.9

* PC: principal component.

**Table 3 molecules-27-02914-t003:** Minimum inhibition concentration of *S. rosmarinus* essential oils tested.

		MIC *v*/*v*						
Cooperative	Strains	$\frac{1}{10}$	$\frac{1}{100}$	$\frac{1}{125}$	$\frac{1}{250}$	$\frac{1}{500}$	$\frac{1}{1000}$	$\frac{1}{2000}$
C1 *	*M. smegmatis*	-	-	-	-	-	+	+
	*E. coli*	-	-	-	-	-	+	+
	*B. subtilis*	-	-	-	-	-	+	+
C2	*M. smegmatis*	-	-	-	-	+	+	+
	*E. coli*	-	-	-	-	+	+	+
	*B. subtilis*	-	-	-	-	-	+	+
C3	*M. smegmatis*	-	-	-	-	-	+	+
	*E. coli*	-	-	-	+	+	+	+
	*B. subtilis*	-	-	-	-	-	+	+

* Cn: Cooperative. +: growth. -: no growth.

**Table 4 molecules-27-02914-t004:** Localization and the coordinates of the samples from the three cooperatives.

Cooperative	Rural Community	Province	Region	Altitude (m)	Latitude N	Longitude W	Maximal Annual Mean Temperature (°C)	Minimal Annual Mean Temperature (°C)	Annual Mean Precipitation (mm)
C1	Talsint	Figuig	Oriental	1 332	32°32′7.8	3°26′22.1	26.0	13.8	140.0
C2	Boumeriem	Figuig	Oriental	1025	33°42′56.2	4°36′5.4	21.3	8.6	184.2
C3	Talsint	Figuig	Oriental	1760	32°38′39.8	3°29′4.6	20.1	7.8	156.9

## Data Availability

All related data are presented within the manuscript.

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
