# Peer review of "Chemometric Investigation and Antimicrobial Activity of Salvia rosmarinus Spenn Essential Oils"

_molecules, 2022, doi:10.3390/molecules27092914_

Round 1
Reviewer 1 Report
Manuscript "Antimicrobial activity and chemometric investigation of the main components and the yield variation of Salvia rosmarinus Spenn essential oils" presents interesting research results. The search for new antimicrobial substances is a very topical topic due to increasing antibiotic resistance.
Detailed comments:
In introduction, it is worth adding to which microorganisms the antimicrobial properties of the tested essentials oils have been demonstrated.
Table 2 - the table is very large and unreadable, it is worth changing its layout to make it more legible.
Why were only 3 types selected for testing antimicrobial properties (please also justify your selection of the microorganisms).
line 406 - missing superscript
line 419 - was the medium with agar for sure?
Why was the MBC not determined after MIC determination?
Author Response
The authors would like to thank the reviewers for the review of our manuscript entitled: “Antimicrobial activity and chemometric investigation of the main components and the yield variation of Salvia rosmarinus Spenn essential oils”. We sincerely appreciate all valuable comments and suggestions, which helped us to improve the quality of the article. Our responses to the Reviewers’ comment are described below in a point-to-point manner. Appropriated changes, suggested by the Reviewers, has been introduced to the manuscript (highlighted within the document).
The responses to reviewers’ comments have been added to the manuscript, the first reviewer with a blue color and the second reviewer with a purple color as well as error corrections in English with a green color (by native English speakers).
Responses to Reviewer 1
Comment 1:
Manuscript "Antimicrobial activity and chemometric investigation of the main components and the yield variation of Salvia rosmarinus Spenn essential oils" presents interesting research results. The search for new antimicrobial substances is a very topical topic due to increasing antibiotic resistance.
Response 1:
The author thank reviewer 1 for his positive comment.
Comment 2:
In introduction, it is worth adding to which microorganisms the antimicrobial properties of the tested essentials oils have been demonstrated.
Response2:
As suggested by the reviewer, the paragraph to which microorganisms the antimicrobial properties of the tested essentials oils have been added in introduction section, line 65 with a new references, namely:
-
10. Borges, R.S.; Ortiz, B.L.S.; Pereira, A.C.M.; Keita, H.; Carvalho, J.C.T. Rosmarinus Officinalis Essential Oil: A Review of Its Phytochemistry, Anti-Inflammatory Activity, and Mechanisms of Action Involved. Ethnopharmacol. 2018, 229, 29–45, doi:10.1016/j.jep.2018.09.038.
-
De Oliveira, J.R.; Camargo, S.E.A.; De Oliveira, L.D. Rosmarinus Officinalis L. (Rosemary) as Therapeutic and Prophylactic Agent. J. Biomed. Sci. 2019, 26.
-
Hamidpour, R. Rosmarinus Officinalis (Rosemary): A Novel Therapeutic Agent for Antioxidant, Antimicrobial, Anticancer,Antidiabetic, Antidepressant, Neuroprotective, Anti-Inflammatory, and Anti-Obesity Treatment. Biomed. J. Sci. Tech. Res. 2017, 1, doi:10.26717/bjstr.2017.01.000371.
-
Sedighi, R.; Zhao, Y.; Yerke, A.; Sang, S. Preventive and Protective Properties of Rosemary (Rosmarinus Officinalis L.) in Obesity and Diabetes Mellitus of Metabolic Disorders: A Brief Review. Curr. Opin. Food Sci. 2015, 2, 58–70.
-
Nieto, G.; Ros, G.; Castillo, J. Antioxidant and Antimicrobial Properties of Rosemary (Rosmarinus Officinalis, L.): A Review. Medicines 2018, 5, 98, doi:10.3390/medicines5030098.
-
Padalia, R.C.; Verma, R.S.; Chauhan, A.; Goswami, P.; Verma, S.K.; Darokar, M.P. Chemical Composition of Melaleuca Linarrifolia Sm. from India: A Potential Source of 1,8-Cineole. Ind. Crops Prod. 2015, 63, 264–268, doi:10.1016/j.indcrop.2014.09.039.
-
Aziz, E.; Batool, R.; Akhtar, W.; Shahzad, T.; Malik, A.; Shah, M.A.; Iqbal, S.; Rauf, A.; Zengin, G.; Bouyahya, A.; et al. Rosemary Species: A Review of Phytochemicals, Bioactivities and Industrial Applications. South African J. Bot. 2021, doi:10.1016/j.sajb.2021.09.026.
-
Bakkali, F.; Averbeck, S.; Averbeck, D.; Idaomar, M. Biological Effects of Essential Oils - A Review. Food Chem. Toxicol. 2008, 46, 446–475.
-
Jiang, Y.; Wu, N.; Fu, Y.J.; Wang, W.; Luo, M.; Zhao, C.J.; Zu, Y.G.; Liu, X.L. Chemical Composition and Antimicrobial Activity of the Essential Oil of Rosemary. Environ. Toxicol. Pharmacol. 2011, 32, 63–68, doi:10.1016/j.etap.2011.03.011.
-
Chraibi, M.; Farah, A.; Elamin, O.; Iraqui, H.; Fikri-Benbrahim, K. Characterization, Antioxidant, Antimycobacterial, Antimicrobial Effcts of Moroccan Rosemary Essential Oil, and Its Synergistic Antimicrobial Potential with Carvacrol. J. Adv. Pharm. Technol. Res. 2020, 11, 25–29, doi:10.4103/japtr.JAPTR_74_19.
Introduction section, line 65: « The essential oil of rosemary is known by its chemical composition which has beneficial properties. It is applied to cure several diseases such as diseases related to inflammation [10], cancer [11], diabetes [12], cardiovascular diseases [13], and Alzheimer’s [14]. It’s used for the treatment of respiratory and inflammatory diseases [15] due to the presence of 1,8-cineole compound. Rosemary essential oil is also renowned by its antimicrobial activity [10,16], while this activity are associated with major chemical compound of essential oil (1,8-cineole and α-pinene) [17]. The 1,8-cineole and α-pinene compounds are known for their antimicrobial activity against some microorganisms, such as Bacillus subtilis (Gram-positive), Escherichia coli (Gram-negative) [18] and Mycobacterium smegmatis (Gram-positive) [19] ».
Comment 3:
Table 2 - the table is very large and unreadable, it is worth changing its layout to make it more legible.
Response 3:
We thank the reviewer for this suggestions. The presentation of Table 2 has been modified to make it more visible and easy to read.
Result and discussion section, variations of the essential oils yield part, line 104:
Table 1. Average yields of S. rosmarinus essential oils of the different samples.
|
Cooperative |
|||||||||
|
Harvest period |
C1* |
C2 |
C3 |
||||||
|
Y1±S.D* |
Y2±S.D |
Average yield |
Y1±S.D |
Y2±S.D |
Average yield |
Y1±S.D |
Y2±S.D |
Average yield |
|
|
P1* |
1.2 ± 0.0 |
1.2 ± 0.1 |
1.2 |
1.1± 0.1 |
1.1 ± 0.0 |
1.1 |
1.2± 0.0 |
1.2± 0.0 |
1.2 |
|
P2 |
2.3 ± 0.0 |
2.4 ± 0.1 |
2.3 |
2.1 ± 0.1 |
2.0 ± 0.0 |
2.0 |
1.7 ± 0.0 |
1.7 ± 0.1 |
1.7 |
|
P3 |
3.1 ± 0.1 |
2.9 ± 0.0 |
3.0 |
2.8 ± 0.0 |
2.7 ± 0.1 |
2.7 |
2.6 ± 0.1 |
2.6 ± 0.1 |
2.6 |
|
P4 |
2.9 ± 0.0 |
2.3 ± 0.1 |
2.6 |
2.3 ± 0.1 |
2.1 ± 0.0 |
2.3 |
2.3 ± 0.1 |
2.1 ± 0.0 |
2.2 |
|
P5 |
2.2 ± 0.0 |
2.0 ± 0.1 |
2.1 |
1.9 ± 0.0 |
2.3 ± 0.0 |
2.1 |
2.2 ± 0.1 |
2.2 ± 0.0 |
2.2 |
|
P6 |
1.7 ± 0.1 |
1.5 ± 0.0 |
1.6 |
1.3 ± 0.0 |
1.3 ± 0.1 |
1.3 |
1.3 ± 0.0 |
1.3 ± 0.0 |
1.3 |
|
P7 |
1.5 ± 0.1 |
1.3 ± 0.0 |
1.4 |
1.2 ± 0.0 |
1.2 ± 0.1 |
1.2 |
1.2 ± 0.0 |
1.3 ± 0.0 |
1.2 |
|
P8 |
1.2 ± 0.1 |
1.2 ± 0.1 |
1.2 |
1.2 ± 0.0 |
1.2 ± 0.1 |
1.2 |
1.1 ± 0.1 |
1.2 ± 0.1 |
1.1 |
|
P9 |
3.2 ± 0.1 |
2.9 ± 0.1 |
3.1 |
2.9 ± 0.1 |
2.8 ± 0.1 |
2.9 |
2.9 ± 0.2 |
2.6 ± 0.0 |
2.8 |
*Pn: harvest period.*Cn: cooperative. *Yn±S.D: yield± standard deviation.
Comment 4:
Why were only 3 types selected for testing antimicrobial properties (please also justify your selection of the microorganisms).
Response 4:
During this study, we tried to determine the antibacterial effect of S. rosmarinus essential oil against Gram-positive and Gram-negative bacteria, in addition these are bacteria which represent impacts on public health and on Agro-food industry. E. coli causes several diseases in humans such as infection of the urinary tract, bloodstream, respiratory tract, cerebrospinal fluid, peritoneum and also causes pneumonia and diarrhoeal diseases of food origin. Tuberculosis is currently the second-leading cause of death worldwide from a single infectious agent after severe acute respiratory syndrome coronavirus 2 (SARS-CoV-2). We used fast-growing non-pathogenic mycobacterial species to facilitate testing and for safety reasons (Mycobacterium smegmatis). In addition, the Bacillus subtilis microorganism is not considered a human pathogen or toxigenic for humans, but it can contaminate food and cause food poisoning. It is a typical Gram-positive foodborne bacterial pathogen. We can extend this antibacterial study to other ATCC-type bacteria such as Staphylococcus aureus, Salmonella Typhi, Acinetobacter baumannii, etc. We mentioned in the conclusion part that the essential oil antibacterial study could be tested on other bacterial strains.
Conclusion section, line 487: «The use of chemometric tools has the potential to identify the compounds responsible for antibacterial activities and will also make it possible to exploit these results to determine the most active oil fractions against pathogenic strains. In this context, this study could be extended to other bacterial strains of medical and food interest».
Comment 5:
Line 406 - missing superscript.
Response 5:
The missing superscript “108 CFU/mL” has been corrected to “108 CFU/mL”.
Comment 6:
Line 419 - was the medium with agar for sure?
Response 6:
Indeed, during this experiment we add a small amount of agar (0.15%), used as a stabiliser of the oil-water mixture, this small amount will not allow the solidification of the culture medium ''Mueller-Hinton'' but it facilitates solubilisation of the essential oil in the liquid culture medium. the addition of agar (0.15%) has been proven in several references, namely:
-
Clinical and Laboratory Standards Institute, Methods for Dilution Antimicrobial Susceptibility Tests for Bacteria that Grow Aerobically; Approved Standard, 8th edCLSI: Wayne, PA, USA, 2009.
-
Bouhdid, J. Abrini, A. Zhiri, M. J. Espuny, and A. Manresa (2009). Investigation of functional and morphological changes in Pseudomonas aeruginosa and Staphylococcus aureus cells induced by Origanum compactum essential oil. Journal of Applied Microbiology, 106, 1558–1568
-
Ismaili, H., Milella, L., Fkih-Tetouani, S., Ilidrissi, A., Camporese, A., Sosa, S., Altinier, G., Loggia, R.D. et al. (2004). In vivo topical anti-inflammatory and in vitro antioxidant activities of two extracts of Thymus satureioides leaves. J Ethnopharmacol 91, 31–36.
Comment 7:
Why was the MBC not determined after MIC determination?
Response7:
We completely agree with you, the determination of the CMB of the essential oil of S. rosmarinus against bacterial strains is very important. The MBC value was defined as the lowest concentration of antimicrobial agent that can kill >99% of the microorganism population when there is no visible growth in nutrient agar. We have already scheduled to do the CMB tests, but unfortunately, during this period access to the laboratory was prohibited due to the confinement of the COVI-19 epidemic, and this period of confinement lasted a very long time, which prevented us completing the CMB tests. The protocol we have programmed is as follows: From each of the wells that is clear, an aliquot of approximately 10 µL will be taken and subcultures onto nutrient agar. The plates will then be incubated at 37°C for 24 hours. Bacterial growth on agar is observed and the concentration which has a colony count of less than 10a will be taken as the MBC value.
Sincerely
Reviewer 2 Report
1.Due to the poor English language, the sentences are confusing and unclear, starting with the abstract ( for example " the most sensitive strain of essential oils"). Extensive editing of English language and style is required.
2. Unclear title of table 1
3. Remove Table 1 from the Introduction and add to the results and discussion. The table is completely confusing and with repetitions of activities. Reformulate the table either by the activities of the dominant components or sort everything by parts of plants in three segments.
4. In the line 311 : The conclusion should be in the direction of comparing the activity of oils and the share of dominant components in them. Of course, one should always keep in mind the contribution of compounds present in a smaller percentage as well as synergistic activity.
5. antimicrobial activity against NOT on
6. How did you prove MIC values if you did not carry over the part of the culture from wells without visible bacterial growth into wells with liquid substrate or solid agar?
Author Response
The authors would like to thank the reviewers for the review of our manuscript entitled: “Antimicrobial activity and chemometric investigation of the main components and the yield variation of Salvia rosmarinus Spenn essential oils”. We sincerely appreciate all valuable comments and suggestions, which helped us to improve the quality of the article. Our responses to the Reviewers’ comment are described below in a point-to-point manner. Appropriated changes, suggested by the Reviewers, has been introduced to the manuscript (highlighted within the document).
The responses to reviewers’ comments have been added to the manuscript, the first reviewer with a blue color and the second reviewer with a purple color as well as error corrections in English with a green color (by native English speakers).
Reponses to reviewer 2
Comment 1:
Due to the poor English language, the sentences are confusing and unclear, starting with the abstract (for example " the most sensitive strain of essential oils"). Extensive editing of English language and style is required.
Response 1:
All linguistic corrections have been made by a green color. The English of the manuscript has been improved carefully for more clarity as suggested.
English has been revised as requested by a native English-speaking.
Comment 2:
Unclear title of table 1.
Response 2:
As suggested by the reviewer in comment 3, Table 1 has been removed from the introduction section and its references added to the results and discussion section.
Comment 3:
Remove Table 1 from the Introduction and add to the results and discussion. The table is completely confusing and with repetitions of activities. Reformulate the table either by the activities of the dominant components or sort everything by parts of plants in three segments.
Response 3:
We think this is an excellent suggestion. The table 1 has been removed from introduction and added to the results and discussion section. The discussion in the antimicrobial activity section has already been improved globally by adding some references of table 1, namely:
- Fadil, M.; Fikri-Benbrahim, K.; Rachiq, S.; Ihssane, B.; Lebrazi, S.; Chraibi, M.; Haloui, T.; Farah, A. Combined Treatment of Thymus Vulgaris L., Rosmarinus Officinalis L. and Myrtus Communis L. Essential Oils against Salmonella Typhimurium: Optimization of Antibacterial Activity by Mixture Design Methodology. Eur. J. Pharm. Biopharm. 2018, 126, 211–220, doi:10.1016/j.ejpb.2017.06.002.
- Messaoudi Moussii, I.; Nayme, K.; Timinouni, M.; Jamaleddine, J.; Filali, H.; Hakkou, F. Synergistic Antibacterial Effects of Moroccan Artemisia Herba Alba, Lavandula Angustifolia and Rosmarinus Officinalis Essential Oils. Synergy 2019, 10, doi:10.1016/j.synres.2019.100057.
- Alvarez, M. V.; Ortega-Ramirez, L.A.; Silva-Espinoza, B.A.; Gonzalez-Aguilar, G.A.; Ayala-Zavala, J.F. Antimicrobial, Antioxidant, and Sensorial Impacts of Oregano and Rosemary Essential Oils over Broccoli Florets. J. Food Process. Preserv. 2019, 43, doi:10.1111/jfpp.13889.
- Nieto, G.; Ros, G.; Castillo, J. Antioxidant and Antimicrobial Properties of Rosemary (Rosmarinus Officinalis, L.): A Review. Medicines 2018, 5, 98, doi:10.3390/medicines5030098.
- JAFARI-SALES, A.; PASHAZADEH, M. Study of Chemical Composition and Antimicrobial Properties of Rosemary (Rosmarinus Officinalis) Essential Oil on Staphylococcus Aureus and Escherichia Coli in Vitro. Int. J. Life Sci. Biotechnol. 2020, doi:10.38001/ijlsb.693371.
- Rapper, S.L. de; Tankeu, S.Y.; Kamatou, G.; Viljoen, A.; van Vuuren, S. The Use of Chemometric Modelling to Determine Chemical Composition-Antimicrobial Activity Relationships of Essential Oils Used in Respiratory Tract Infections. Fitoterapia 2021, 154, doi:10.1016/j.fitote.2021.105024.
- Kovač, J.; Šimunović, K.; Wu, Z.; Klančnik, A.; Bucar, F.; Zhang, Q.; Možina, S.S. Antibiotic Resistance Modulation and Modes of Action of (-)-α-Pinene in Campylobacter Jejuni. PLoS One 2015, 10, doi:10.1371/journal.pone.0122871.
- Ojeda-Sana, A.M.; van Baren, C.M.; Elechosa, M.A.; Juárez, M.A.; Moreno, S. New Insights into Antibacterial and Antioxidant Activities of Rosemary Essential Oils and Their Main Components. Food Control 2013, 31, 189–195, doi:10.1016/j.foodcont.2012.09.022.
- Rapper, S.L. de; Tankeu, S.Y.; Kamatou, G.; Viljoen, A.; van Vuuren, S. The Use of Chemometric Modelling to Determine Chemical Composition-Antimicrobial Activity Relationships of Essential Oils Used in Respiratory Tract Infections. Fitoterapia 2021, 154, doi:10.1016/j.fitote.2021.105024.
- Miladinović, D.L.; Dimitrijević, M. V.; Mihajilov-Krstev, T.M.; Marković, M.S.; Ćirić, V.M. The Significance of Minor Components on the Antibacterial Activity of Essential Oil via Chemometrics. LWT 2021, 136, doi:10.1016/j.lwt.2020.110305.
Results and discussion section, antimicrobial activity part, line 338 : « Fadil et al. [61] reported that rosemary essential oil represented an antimicrobial activity against Salmonella typhimurium with a MIC value of 2% (v/v).The MIC value of rosemary essential oil found by Messaoudi Moussii et al. [62] against three bacterial strains (S.aureus, E.coli and Pseudomonas aeruginosa) was 3.33, 1.67, and 6.67 µL/mL, respectively. According to Alvarez et al. [63] the rosemary essential oil showed an antimicrobial activity against E. coli, Salmonella choleraesuis, Listeria monocytogenes, and S. aureus with a MIC value of 0.65, 0.65, 0.91, and 0.91 mg/mL, respectively ».
Results and discussion section, antimicrobial activity part, line 357: « Several authors demonstrated that the chemical composition has an effect on antimicrobial activity such as 1,8-cineole and α-pinene [64,65]. According to Rapper et al. [66] the antibacterial activity against M. smegmatis of rosemary was most likely to occur when high quantities of 1,8-cineole were presented. Moreover, Kovač et al. [67] demonstrated that the α-pinene compound presented an antimicrobial activity against 16 strains of Campylobacter jejuni. Ojeda-Sana et al. [68] indicated that the rosemary essential oil has a high antibacterial activity against the Gram-positive bacteria (S. aureus and Enterococcus faecalis), and against the Gram-negative bacteria (E. coli and K. pneumoniae). They also demonstrated that the α-pinene compound presented a broad antibacterial spectrum against Gram-negative and Gram-positive bacteria, as well as the 1,8-cineole compound showed antibacterial activity against pathogenic bacteria by causing a rupture of the cell membrane. It is known by the scientific community that the main compounds of essential oils have a key role in antibacterial activity. However, the importance of minor components or a combination thereof could explain the complete activity of essential oils against clinical and foodborne pathogens [69,70] ».
Comment 4:
In the line 311: The conclusion should be in the direction of comparing the activity of oils and the share of dominant components in them. Of course, one should always keep in mind the contribution of compounds present in a smaller percentage as well as synergistic activity.
Response 4:
We thank the reviewer for this viewpoint and we agree. The conclusion of the antimicrobial activity section has been enhanced by the addition of comparing the activity of oils and the share of dominant components in them. These modifications have been noted as suggested by reviewer in results and discussion section, antimicrobial activity part, line 349.
Results and discussion section, antimicrobial activity part, line 349: « It can be concluded that the rosemary essential oil from cooperative C1 exhibited the best antimicrobial activity of the three cooperatives. In fact, this can be explained by a high amount of 1,8-cineole compound in rosemary essential oil from C1 (51.1±1.8%) compared to two cooperatives C2 and C3 (46.2±0.3 and 47.3±0.8%, respectively), but the contributions of α–pinene and camphor compounds should also be noted. In addition, the analysis of the S. rosmarinus essential oil belonging to the cooperative N° 1 showed a high rate of camphor (7.3 ± 0.5%) compared to the two other cooperatives C2 (5.8±0.2%) and C3 (5.3±0.1%). The essential oil from the cooperative C2 represented the highest amount of the α-pinene compound (11.6±0.5%), followed by C3 (10.7±0.5%) and then C1 with a value of 9.9±0.2%. The effect of synergy could also be the origin of this activity. Several authors demonstrated that the chemical composition has an effect on antimicrobial activity such as 1,8-cineole and α-pinene [64,65]. According to Rapper et al. [66] the antibacterial activity against M. smegmatis of rosemary was most likely to occur when high quantities of 1,8-cineole were presented. Moreover, Kovač et al. [67] demonstrated that the α-pinene compound presented an antimicrobial activity against 16 strains of Campylobacter jejuni. Ojeda-Sana et al. [68] indicated that the rosemary essential oil has a high antibacterial activity against the Gram-positive bacteria (S. aureus and Enterococcus faecalis), and against the Gram-negative bacteria (E. coli and K. pneumoniae). They also demonstrated that the α-pinene compound presented a broad antibacterial spectrum against Gram-negative and Gram-positive bacteria, as well as the 1,8-cineole compound showed antibacterial activity against pathogenic bacteria by causing a rupture of the cell membrane. It is known by the scientific community that the main compounds of essential oils have a key role in antibacterial activity. However, the importance of minor components or a combination thereof could explain the complete activity of essential oils against clinical and foodborne pathogens [69,70] ».
Comment 5:
Antimicrobial activity against NOT on.
Response 5:
The word “on” has been replaced by “against” in the sentence.
Comment 6:
How did you prove MIC values if you did not carry over the part of the culture from wells without visible bacterial growth into wells with liquid substrate or solid agar?
Response 6:
Thank you very much for this remark, the determination of the MICs is based on the color change of the samples by adding the rezasurin. The presence of red color in the well indicated bacterial growth while no color change in the well indicated no growth. The MIC value was defined as the lowest antimicrobial concentration preventing visible bacterial growth. The figure above, illustrates an example of our results for the determination of the MICs of the essential oil of S. rosmarinus against E. coli ATCC ATCC25922 and Bacillus subtilis ATCC 23857. This clarification has been added in the material and methods section, antimicrobial activity part (minimal inhibitory concentration, line 454) to make the technique more explicit.
Materials and Methods section, antimicrobial activity part (minimal inhibitory concentration), line 454: « The MIC was presented as the lowest essential oil concentration that showed a negative bacterial growth translated by a non-change in rezasurin color. A positive one is detected by a reduction of blue dye resazurin to pink resorufin».
Thank you again for your comments that permitted to increase the quality of our paper. We hope that our answers will suit the reviewers’ expectations.
Sincerely